# The Impact of Rubella Vaccine Introduction on Rubella Infection and Congenital Rubella Syndrome: A Systematic Review of Mathematical Modelling Studies

**DOI:** 10.3390/vaccines9020084

**Published:** 2021-01-25

**Authors:** Nkengafac Villyen Motaze, Zinhle E. Mthombothi, Olatunji Adetokunboh, C. Marijn Hazelbag, Enrique M. Saldarriaga, Lawrence Mbuagbaw, Charles Shey Wiysonge

**Affiliations:** 1National Institute for Communicable Diseases (NICD), A Division of the National Health Laboratory Service (NHLS), Johannesburg 2131, South Africa; 2Division of Epidemiology and Biostatistics, Department of Global Health, Stellenbosch University, Cape Town 7505, South Africa; olatunji@sun.ac.za (O.A.); mbuagblc@mcmaster.ca (L.M.); Charles.Wiysonge@mrc.ac.za (C.S.W.); 3Centre for the Development of Best Practices in Health (CDBPH), Yaoundé Central Hospital, Yaoundé 1211, Cameroon; 4The South African Department of Science and Innovation-National Research Foundation (DSI-NRF), Centre of Excellence in Epidemiological Modelling and Analysis (SACEMA), Stellenbosch 7600, South Africa; zinhlem@sun.ac.za (Z.E.M.); marijnhazelbag@sun.ac.za (C.M.H.); 5The Comparative Health Outcomes, Policy, and Economics (CHOICE) Institute, University of Washington, Seattle, WA 98195, USA; emsb@uw.edu; 6Department of Health Research Methods, Evidence, and Impact (HEI), McMaster University, Hamilton, ON L8S 4L8, Canada; 7Biostatistics Unit, The Research Institute, St Joseph’s Healthcare, Hamilton, ON L8N 4A6, Canada; 8Cochrane South Africa, South African Medical Research Council, Cape Town 7505, South Africa; 9School of Public Health and Family Medicine, University of Cape Town, Anzio Road, Observatory, Cape Town 7935, South Africa

**Keywords:** rubella, congenital rubella syndrome, rubella-containing vaccines, systematic review, data synthesis

## Abstract

Introduction: Rubella vaccines have been used to prevent rubella and congenital rubella syndrome (CRS) in several World Health Organization (WHO) regions. Mathematical modelling studies have simulated introduction of rubella-containing vaccines (RCVs), and their results have been used to inform rubella introduction strategies in several countries. This systematic review aimed to synthesize the evidence from mathematical models regarding the impact of introducing RCVs. Methods: We registered the review in the international prospective register of systematic reviews (PROSPERO) with registration number CRD42020192638. Systematic review methods for classical epidemiological studies and reporting guidelines were followed as far as possible. A comprehensive search strategy was used to identify published and unpublished studies with no language restrictions. We included deterministic and stochastic models that simulated RCV introduction into the public sector vaccination schedule, with a time horizon of at least five years. Models focused only on estimating epidemiological parameters were excluded. Outcomes of interest were time to rubella and CRS elimination, trends in incidence of rubella and CRS, number of vaccinated individuals per CRS case averted, and cost-effectiveness of vaccine introduction strategies. The methodological quality of included studies was assessed using a modified risk of bias tool, and a qualitative narrative was provided, given that data synthesis was not feasible. Results: Seven studies were included from a total of 1393 records retrieved. The methodological quality was scored high for six studies and very high for one study. Quantitative data synthesis was not possible, because only one study reported point estimates and uncertainty intervals for the outcomes. All seven included studies presented trends in rubella incidence, six studies reported trends in CRS incidence, two studies reported the number vaccinated individuals per CRS case averted, and two studies reported an economic evaluation measure. Time to CRS elimination and time to rubella elimination were not reported by any of the included studies. Reported trends in CRS incidence showed elimination within five years of RCV introduction with scenarios involving mass vaccination of older children in addition to routine infant vaccination. CRS incidence was higher with RCV introduction than without RCV when public vaccine coverage was lower than 50% or only private sector vaccination was implemented. Although vaccination of children at a given age achieved slower declines in CRS incidence compared to mass campaigns targeting a wide age range, this approach resulted in the lowest number of vaccinated individuals per CRS case averted. Conclusion and recommendations: We were unable to conduct data synthesis of included studies due to discrepancies in outcome reporting. However, qualitative assessment of results of individual studies suggests that vaccination of infants should be combined with vaccination of older children to achieve rapid elimination of CRS. Better outcomes are obtained when rubella vaccination is introduced into public vaccination schedules at coverage figures of 80%, as recommended by WHO, or higher. Guidelines for reporting of outcomes in mathematical modelling studies and the conduct of systematic reviews of mathematical modelling studies are required.

## 1. Introduction

Rubella-containing vaccines (RCVs) were first introduced in Europe and the USA in 1969 [1], resulting in a decline in the number of rubella infections and cases of congenital rubella syndrome (CRS). Rubella causes mild disease in most children and adults. Severe complications of rubella infections occur mostly in pregnant women, and these include miscarriages, stillbirths, and CRS. CRS can occur in up to 90% of cases when a pregnant woman gets infected with rubella in the first trimester [2]. Therefore, rubella vaccines not only prevent infection in children and adults, but also indirectly protect the foetus. 

The global vaccine action plan (GVAP) [3] and the global measles and rubella strategic plan resulted in the establishment of measles and rubella elimination targets for several World Health Organization (WHO) regions [4]. There was subsequently an accelerated roll-out of RCV into the public immunization schedules of countries that did not include rubella vaccination in their national immunization programs. By the end of 2019, only 21 countries did not include RCVs in their public immunization schedules [5]. Different RCV introduction strategies and their impact on rubella and CRS elimination have been outlined by WHO [6]. These included childhood vaccination only or various combinations of childhood and adult vaccination. 

A systematic review (SR) makes use of predetermined methods to obtain, evaluate, and collate available individual studies on a specific research question [7,8]. SRs are deemed as providing the highest level of evidence for health care interventions [9]. When assessing individual studies, comparisons can be made to identify similarities or differences in study characteristics (e.g., setting, design, participants, etc.), which influence the applicability of the results to different settings. Meta-analysis (which involves combining results of several individual studies), when appropriate, allows for greater precision, since the resulting sample size is larger than that of individual studies.

Five main steps have been suggested for evidence-based practice [10] and SRs [11]. These steps have been widely adopted by researchers to address a variety of research questions. However, the study question of interest determines the design of individual studies that are included. SRs of classical epidemiological studies have been rigorously improved over time with published methodological [8] and reporting approaches [12,13] that are regularly updated. In contrast, although there have been several published SRs of mathematical modelling studies, guidelines for their design and implementation have not yet been extensively developed. Examples of study questions addressed by SRs of mathematical modelling include interventions on health care provision in small and large populations [14] and the impact of vaccines on tuberculosis [15] and cervical cancer [16].

Applications of mathematical modelling studies to vaccination strategies are broad. Identifying barriers to achieving elimination of measles [17] and informing vaccine introduction into national public immunization programs [15,18,19,20] are a few examples. Mathematical modelling studies are referred to as dynamical or mechanistic epidemiological studies, while observational and interventional studies (such as cross-sectional, case-control, cohort, and randomized controlled trials) are referred to as classical epidemiological studies [21]. A mathematical model uses mathematical statements to represent observations [22]. In general, when the model output solely depends on the inputs, the model is said to be deterministic, and when the role of chance is incorporated into the model, the model is said to be stochastic. Using computer software, a mathematical model can be used to simulate or represent a biological process, and with advances in technology, there have been advances in the understanding of complex disease processes. Models of rubella transmission dynamics build on current knowledge of pathogen biology and separate the population into compartments depending on disease stage or vaccination status. These compartments could be individuals with maternal immunity (M), exposed individuals who are infected but not yet infectious (E), infected individuals who are infectious (I), previously infected but recovered individuals (R), and vaccinated individuals (V).

In contrast to classical epidemiological studies that deduce conclusions on data collected in the real world, mathematical modelling studies can simulate interventions and estimate their impact. Insights on interventions that would be challenging to implement, such as testing various vaccine introduction scenarios in a given country, can be obtained. Currently, the synthesis of evidence from mathematical modelling studies has not been as extensive as that of classical epidemiology studies. However, methodically compiling evidence can yield valid findings to inform policy.

Mathematical models have been used to assess the impact of RCV on rubella and CRS elimination in several WHO regions: Costa Rica [23] in the Americas, India [19] in South East Asia, and Madagascar [18] and South Africa [20] in Africa. Given the variety of settings and modelling approaches used to evaluate the impact of RCVs, it is important to comprehensively summarize the evidence to inform policy-makers in countries that have not yet introduced RCVs, or guide adjustment of vaccination strategies where RCV are already being used. 

This study aimed to summarize the evidence from mathematical modelling on RCV introduction scenarios and their impact on rubella transmission dynamics. The primary objective was to estimate time to CRS elimination following rubella vaccine introduction. Secondary objectives were to describe the main modelling approaches to rubella vaccine introduction strategies, identify vaccine introduction strategies that achieve the most rapid reduction in cases of rubella and CRS, and outline the most cost-effective vaccine introduction strategies.

## 2. Materials and Methods 

This systematic review follows the Preferred Reporting Items for Systematic Reviews and Meta-Analyses (PRISMA) statement [12]. Given that the PRISMA statement was developed mainly for classical epidemiological studies, we adapted the items of the PRISMA checklist where applicable.

### 2.1. Inclusion Criteria

Study design: We included mechanistic or predictive mathematical modelling studies that simulate rubella vaccine introduction into national immunization schedules. We included both deterministic and stochastic models. Scenarios of interest targeted various population age groups and scheduling of vaccination, i.e., combining routine doses and mass campaigns.
*Participants*: Individuals eligible for rubella vaccination of any age in any country*Intervention*: Rubella-containing vaccine introduction scenarios*Comparison*: No rubella vaccine or different vaccine introduction scenarios*Outcomes*: We included studies that reported at least one of the following outcomes of RCVs at a population level: time (in years) to the elimination of CRS, time (in years) to rubella elimination, description of trends in rubella and CRS incidence, number of vaccinated individuals per CRS case averted, and cost-effectiveness of vaccine introduction strategy.*Time horizon*: We included studies in which the time horizon from the year of vaccine introduction to the end of the simulation is at least five years. We assumed it is unlikely that any meaningful impact of rubella vaccine introduction will be measurable within a shorter period.

### 2.2. Exclusion Criteria

We excluded epidemiological studies with an interventional or observational design. We also excluded mathematical modelling studies that were focused on the estimation of model parameters (e.g., basic reproductive number) and modelling studies in which additional vaccination strategies were tested in a setting that already had public sector vaccination. 

### 2.3. Search Strategy

A comprehensive search strategy was developed and implemented to obtain published studies in Medline and Scopus. Different combinations of Medical Subject Heading (Mesh) terms were used to maximize the outputs of the electronic search. We reviewed the references of included studies for other potentially eligible studies. We also searched for unpublished studies from conference abstracts and repositories of student theses. We only included studies published between 1 January 2000, and 20 June 2020 (to cover a period of 20 years), and we did not apply any language restrictions. The search strategy is presented in Appendix A.

### 2.4. Study Selection and Data Extraction

Two authors independently reviewed the abstracts of studies retrieved using the search strategy. When the abstracts suggested that the studies met inclusion criteria, full-text articles were reviewed to make a final decision. A data collection tool was developed to extract information on study characteristics, risk of bias, and participant, intervention, comparison, outcome, and time horizon (PICOT) items from the included studies. 

### 2.5. Risk of Bias Assessment

We assessed the methodological quality of included studies using the risk of bias tool used in previously published studies [14,15,24] (Appendix A). This risk of bias tool includes questions on the following criteria: study aims and objectives, population and setting, intervention and comparator, outcome measures, time horizon, modelling methods, parameter ranges and sources, assumptions, uncertainty analyses, model fitting, model validation, presentation of results, discussion, and conflict of interest.

The 14 risk of bias criteria consisted of one or more questions addressing specific aspects of the study and were graded as poor (if score = zero), average (if score = one), and good (if score = two). The allocated score for each risk of bias criterion was two if all responses to the questions were “yes”, one if at least one of the responses was “yes”, and zero if none of the responses was “yes”. The scores were added to obtain an overall risk of bias score ranging from zero to 28 for the given study. Based on this score, the methodological quality of included studies was classified as very high (score > 22), high (19–22), medium (14–18), and low (<14). 

### 2.6. Data Analysis

We performed a qualitative synthesis of the included studies. The minimum WHO-recommended coverage of RCV is 80% [15], so we used this value as the basis for comparing outcomes reported by different studies in cases where several vaccine coverage values were simulated. If any study did not report outcomes for 80% vaccine coverage, outcome values for the next highest coverage values closest to 80% were reported.

We had planned to derive random-effects pooled predictions of the population-level impact of RCV using the metaphor package in R statistical software version 4.0 [25]. We intended to assign equal weights to all models and estimate the median (along with 10th, 25th, 75th, and 90th percentiles) time to congenital rubella syndrome elimination. We had planned to use univariable and/or multivariable linear meta-regression (depending on the number of included studies) to identify potential sources of heterogeneity among included studies and conduct subgroup analysis for different groups of models (deterministic versus stochastic), different World Health Organization Regions, and World Bank country classifications. However, differences in the reporting of outcomes between individual studies did not allow for pooled estimates to be obtained.

### 2.7. Ethical Considerations

This systematic review did not include the use of individual participant data. Therefore, we did not seek ethical approval. In line with PRISMA recommendations, the study proposal was registered with PROSPERO (CRD42020192638), an international prospective register of systematic reviews, before conducting the search. The PRISMA checklist is included in Appendix A. 

## 3. Results

The search strategy retrieved 1393 records, and 561 distinct abstracts were assessed for inclusion. We excluded 537 records based on the abstracts and reviewed 24 full text articles. We excluded a further 17 articles [20,23,26,27,28,29,30,31,32,33,34,35,36,37,38,39,40] and included seven studies in the review (see Figure 1 and Appendix A). Only a narrative synthesis was done, because several included studies did not include uncertainty intervals when reporting outcomes.

### 3.1. Included Studies

#### 3.1.1. Characteristics of Included Studies

Seven studies were included and all studies implemented age-structured deterministic models. Two studies were conducted in Africa [18,41], one in Europe [42], and four in Asia [19,43,44,45]. According to the World Bank classification of countries [46], three studies simulated RCV introduction in lower-middle income countries (India, Madagascar, and Vietnam) [18,19,44], three in upper-middle income countries (China, Indonesia, and South Africa) [41,43,45], and one in a high-income country (Croatia) [42]. Regarding models’ compartments, three studies used MSIRV [18,19,41], three studies used MSEIRV [42,43,44], and one study used SEIRV [45]. The number of scenarios simulated ranged from three to eight. The characteristics of these included studies are shown in Table 1.

#### 3.1.2. Risk of Bias in Included Studies

None of the included studies was classified as low or medium with respect to methodologic quality. The methodological quality was high for six studies and very high for one study. The two risk of bias criteria with the lowest scores were method of fitting and model validation. None of the included studies reported the method of fitting, and only one study described the validation method used. The highest scores were assigned for aim and objectives, setting and population, intervention and comparators, outcome measures, model structure and time horizon, modelling methods, and assumptions. Methodological quality assessments by study and by risk of bias criterion are shown in Figure 2.

### 3.2. Effect of Rubella Vaccine Introduction

The included studies simulated a variety of vaccine introduction scenarios involving routine infant vaccination and mass campaigns. Some scenarios were simulated by several studies, while others were unique to individual studies. Five studies [18,19,41,43,44] simulated national RCV introduction, while two studies [42,45] simulated introduction in a limited area within a country. Among the studies that simulated nation-wide vaccine introduction, two studies [18,19] mostly reported outcomes at sub-national level.

#### 3.2.1. Time to Elimination of CRS

None of the included studies reported the time from introduction of RCVs to elimination of CRS.

#### 3.2.2. Trends in CRS Incidence

Six studies reported on changes in CRS incidence following introduction of RCVs. The number of years to CRS elimination was not specified in the studies, but this outcome could be extrapolated from the data on incidence trends within five-year intervals.

Gao et al. found that compared to no RCV, routine vaccination of one-year-olds resulted in higher CRS incidence at vaccine coverage figures ≤50%. When RCV coverage was ≤70%, CRS incidence was lower with RCVs. At 90% vaccine coverage, rubella elimination was achieved over a period between 15–20 years. When comparing vaccination of 12-year-old girls to no RCV, CRS incidence was lower at all simulated vaccine coverage levels. CRS elimination was not achieved, even with 90% vaccine coverage. Trends in CRS incidence were not described for other vaccine introduction scenarios.

Motaze et al. found that CRS incidence was lower for all vaccine introduction scenarios relative to no RCV for all levels of vaccine coverage simulated (60–95%). With routine vaccination of one-year-olds, CRS elimination was achieved in 15–20 years at RCV coverage ≤80%, and with routine vaccination of one-year-olds combined with nine-year-olds, CRS elimination was achieved in 5–10 years. CRS elimination was achieved for all scenarios involving routine vaccination of one-year-olds combined with mass vaccination of 1–14-year olds and/or 1–4-year-olds at vaccine coverage figures ≤65% in 0–5 years.

Vynnycky et al. simulated RCV introduction at a fixed coverage level of 90% and found that CRS elimination was achieved in under five years for all scenarios involving routine vaccination of nine-month-olds combined with either mass vaccination of nine months–14 year olds or 15–35-year-old females.

Wesolowski et al. also simulated RCV introduction at a fixed coverage level, but coverage levels differed by region. The incidence of CRS was lower for all vaccine introduction scenarios compared to no RCV. For scenarios involving combinations of routine vaccination and mass campaigns, the effects of mass campaigns targeting individuals above 10 years of age do not differ from targeting 10-year-old children.

Winter et al. performed simulations with different RCV coverage for various regions. Compared to no RCV, private sector vaccination of children at 9–15 months and 4–6 years resulted in higher CRS incidence compared to no RCV. CRS incidence was lower with routine vaccination of children aged 9–12 and 16–24 months old (RCV coverage = 60%) combined with a mass campaign targeting children aged nine months through 14 years (RCV coverage = 60%) than with private sector vaccination in all regions at R0 = 5. With higher values of R0 (7, 9, and 11), CRS incidence was higher compared to private sector vaccination in several regions. With routine vaccination coverage of 80% targeting children aged 9–12 and 16–24 months old combined with a mass campaign with 80% vaccine coverage targeting children aged nine months through 14 years, CRS incidence was lower than with private sector vaccination irrespective of R0 values.

Wu et al. reported that incidence of CRS was lower for all vaccine introduction scenarios compared to when RCVs were not included in the public vaccination schedule. CRS elimination was achieved only in scenario 6.

#### 3.2.3. Time to Elimination of Rubella

None of the included studies reported the time from RCV introduction to rubella elimination.

#### 3.2.4. Trends in Rubella Incidence

Wesolowski et al. and Vynnycky et al. did not report rubella incidence over time following RCV introduction. Gao et al., Jazbec et al., Motaze et al., and Wu et al. reported lower rubella incidence for all vaccine introduction scenarios compared to no RCV introduction. The drop in rubella incidence was abrupt in scenarios including a mass campaign while rubella incidence dropped progressively (over five to 10 years) for scenarios that did not include a mass campaign. Winter et al. found that rubella incidence remained below 5/100,000 live births for the entire duration of the simulation when routine vaccination coverage was above 95%.

#### 3.2.5. Number of Vaccinated Individuals per CRS Case Averted

Gao et al. reported the lowest number of vaccine doses per CRS case averted 46 years after RCV introduction in scenario 2. Vaccine doses per CRS case averted in each scenario were as follows: scenario 1 = 1500, scenario 2 = 1421, scenario 3 = 1439, scenario 4 = 4474, scenario 5 = 6403, scenario 6 = 2622, and scenario 7 = 2329. Motaze et al. reported that at 80% coverage, the lowest number of vaccine doses per CRS case averted 20 years after RCV introduction was achieved with scenario 6. 

### 3.3. Economic Evaluation Measure

#### 3.3.1. Disability-Adjusted Life Years (DALYs) Averted

Only one study, Motaze et al., reported this outcome. At 80% RCV coverage, undiscounted DALYs averted 20 years after vaccine introduction was the same for scenarios 3, 4, and 5 (178,584). DALYs averted were lowest for scenario 2 (138,408), followed by scenario 6 = 168,562. 

#### 3.3.2. Vaccine Cost per CRS Cases Averted

Two studies reported outcomes related to vaccine cost. Motaze et al. reported that at 80% coverage, the lowest cost per CRS case averted 20 years after RCV introduction was achieved with scenario 6. Wu et al. found that at 80% coverage, the lowest discounted incremental cost-effectiveness ratio (ICER) post vaccine introduction (cost per CRS case averted 20 years after RCV introduction) was obtained with scenario 2 (USD 277.22). This cost was USD 375.22 for scenario 1, USD 440.15 for scenario 3, USD 571.33 for scenario 4, USD 761.65 for scenario 5, USD 1098.29 for scenario 6, and USD 739.93 for scenario 7. 

None of the studies reported time to CRS elimination and time to rubella elimination.

## 4. Discussion

All the studies included in this review were deterministic age-structured models. The differences in the results of individual studies did not allow for data synthesis. CRS elimination was achieved over the shortest period with scenarios combining routine immunization of infants to mass vaccination of older individuals. Low coverage with rubella vaccines led to higher CRS incidence compared to no vaccination. Interestingly, studies that assessed cost implications found that strategies involving routine vaccination of children at specific ages outside the routine infant dose were more cost effective than strategies involving mass campaigns.

The reported increase in CRS incidence following rubella vaccination at low coverage is a well described phenomenon [2,47]. The low vaccine coverage resulted from low public vaccination coverage in one study and private sector vaccination in another. It is unlikely for private sector vaccination against rubella to be adopted as a national strategy, but this highlights the dangers of not having a public rubella vaccination policy. Public sector vaccination, despite being the obligatory strategy to be included for achieving rubella and CRS elimination, can achieve low levels of coverage if not properly implemented. Achieving at least 80% coverage as recommended by WHO [16] avoids this negative effect of rubella vaccines, and countries planning RCV introduction should adhere to this recommendation. 

None of the studies reported time to rubella elimination or time to CRS elimination, which are critical outcomes that are important to policy-makers. Arriving at a point estimate is standard in classical epidemiology, but dynamical epidemiological studies focus more on understanding factors driving the disease transmission process. Elimination of CRS can be achieved using RCV. Given that countries and funders face competing priorities, optimal vaccination strategies have to be chosen when planning RCV introduction. Routine infant vaccination during the first year of life coupled with vaccination of older children and/or adults was found to result in elimination of CRS over a shorter period. 

Comparing the costs or vaccination to the benefits is important in order to maximize use of available resources. None of the included studies were economic evaluation models, but vaccinating children at a specific age within the first decade of life was associated with the lowest cost per CRS case averted. Without public vaccination against rubella, infections predominantly affect children 5–14 years old [48], which could explain why strategies targeting children in this age group lead to more rapid elimination. Furthermore, the drop in rubella incidence with vaccination of children at six or nine years is slower compared to the rapid drop with mass campaigns, with similar long-term impact at lower costs. Several countries previously implemented strategies that do not include a mass campaign but targeted vaccination of older individuals who are not in the age group for routine infant vaccination [49,50]. Subsequently, the insufficient decrease in rubella transmission allowed for persistence of CRS, leading to modifications in vaccination strategy to incorporate additional vaccination of infants.

All studies that reported rubella incidence found a reduction in incidence for all vaccine coverage levels in all scenarios. While rubella incidence reduces following vaccination, the changes in transmission dynamics with corresponding increase in average age of infection when only infants are vaccinated could have an undesirable impact on CRS incidence. Trends in rubella and CRS incidence were extracted from the included studies, and approximate periods required to achieve elimination were approximated. This is not an accurate method of obtaining estimates, and the fact that individual studies presented their results in such a manner means it is possible to improve reporting of disease incidence. 

There are several tools available for assessing the methodological quality of mathematical modelling studies. The risk of bias tool used in our study was developed building several existing tools. The systematic review approach for mathematical modelling studies is still being developed, including risk of bias assessment. Several items of the tool we used combined multiple aspects of the study design, which makes it difficult to clearly tease out the implications of each of these aspects on the overall quality assessment. Several studies did not report on the method of model validation, and none of the included studies reported on the method of fitting. Recommendations for validation of mathematical models have been proposed [51], but there is no consensus on a preferred method. It is common for a validated, published model to be used for simulating disease dynamics in different settings. In this case, authors could refer to the original published model but do not provide details on the method of model validation or fitting. Bearing in mind that it could be laborious to repeat previously published information, it is helpful for researchers conducting systematic reviews to have a clear understanding of the model without having to search further. 

Differences in the manner in which outcomes were reported by different studies, even for outcomes reported by more than one included study, rendered quantitative data synthesis impossible. Authors of individual studies could have reported different outcomes considered to be relevant for the settings in which RCV were simulated. However, the lack of recommended outcomes does not encourage authors to report results in such a way that allows for meta-analyses. This limits the ability for more precise estimates to be obtained from individual mathematical modelling studies. 

The main limitation of this study relates to differences between included studies in terms of reported outcomes and scenarios simulated that did not allow for synthesis of results. The main strength of the study is that the included studies each modelled rubella vaccine introduction in different settings. This implies that the findings are applicable to those settings and could better inform decision-making.

## 5. Conclusions

We were unable to conduct data synthesis of included studies due to discrepancies in outcome reporting. However, qualitative assessment of results of individual studies suggests that, in addition to infant vaccination, countries introducing rubella vaccination in their EPI schedule should vaccinate older children and/or adults in order to achieve more rapid decreases in rubella and CRS incidence. There is a wide variety of possible scenarios available to policy-makers in countries that do not yet include rubella vaccination in their public vaccination schedules, but irrespective of the vaccine introduction strategy chosen, improved outcomes were obtained for coverage figures of 80% (the minimum WHO-recommended coverage) and above.

Researchers modelling rubella vaccine introduction should attempt to report effect estimates and corresponding uncertainty intervals to enable pooling of results. Guidelines on reporting of individual mathematical modelling studies and systematic reviews of mathematical modelling studies should be developed such that evidence from mathematical modelling studies can be summarized in a consistent and structured manner.

## Figures and Tables

**Figure 1 vaccines-09-00084-f001:**
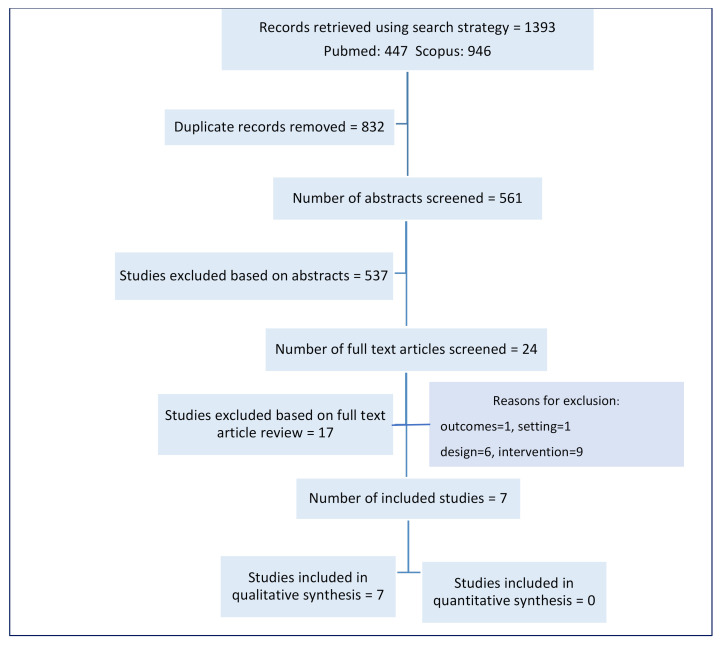
Study flow diagram.

**Figure 2 vaccines-09-00084-f002:**
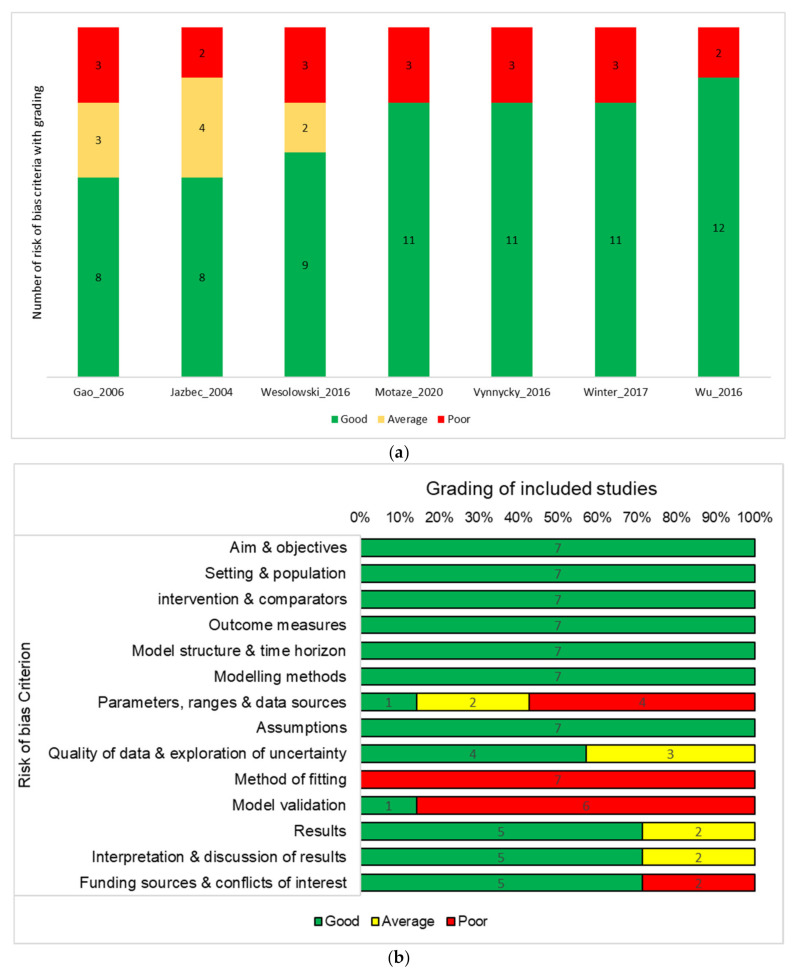
(**a**) Each of the 14 risk of bias items is allocated a score of 0 (poor), 1 (average) or 2 (good). The number of items in each category is presented for each included study; (**b**) Each of the 14 risk of bias items is allocated a score of 0 (poor), 1 (average), or 2 (good). The number of studies that had the a given scoring for each item is presented. Risk of bias assessment by study (**a**) and by risk of bias criterion (**b**).

**Table 1 vaccines-09-00084-t001:** Characteristics of included studies.

Study	Description of Target Age Groups and Sex for Each Vaccine Introduction Scenario	Setting	WHO Region	World Bank Grading	Previous Private Sector RCV	Time Frame	Classes	Reported Outcomes
Gao_2016 [43]	- scenario 1: routine vaccination at 1 year (M & F); - scenario 2: mass vaccination of 2–14-year-olds (F) and routine 12-year-olds (F); - scenario 3: mass vaccination of 2–14-year-olds (F); - scenario 4: mass vaccination of 2–14-year-olds (M & F); - scenario 5: mass vaccination of 15–40-year-olds (F); - scenario 6: routine vaccination of 1-year-old children (M & F) and mass vaccination of 2–14-year-olds (M & F) and 15–40-year-olds (F); - scenario 7: routine 1-year-old children (M & F), mass vaccination of 2–14-year-old girls and mass 15–40-year-old women; - scenario 8: routine 12-year-olds (F).	China	Western Pacific	Upper-middle Income	Yes	46 years	MSEIRV	- Trends in rubella incidence - Trends in CRS incidence - Number of vaccinated individuals - Number of vaccinated individuals per CRS case averted
Jazbec_2004 [42]	- scenario 1: routine vaccination at 1 year (M & F) and at 14 (F); - scenario 2: routine vaccination at 1 + 7 years (M & F), and at 14 years (F); - scenario 3: routine vaccination at 1 + 12 years (M & F).	Croatia, Tresnjevka municipality	Europe	High-income	No	55 years	MSEIRV	- Trends in rubella incidence
Motaze_2020 [41]	- scenario 1: private vaccination only (M & F); - scenario 2: private + routine vaccination at 1 year (M & F).; - scenario 3: private + routine vaccination at 1 year and start-up campaign for 1–14 year-olds (M & F); - scenario 4: private + routine vaccination at 1 year and start-up campaign for 1–14 years, followed by one follow-up campaign for 1–4 year-olds (M & F); - scenario 5: private + routine vaccination at 1 year and start-up campaign for 1–14 year-olds, followed by follow-up campaigns every 5 years for 1–4 year-olds (M & F); - scenario 6: private + routine vaccination targeting 1 year and routine vaccination for 9-year-olds (M & F).	South Africa	Africa	Upper-middle income	Yes	30 years	MSIRV	- Trends in rubella incidence - Trends in CRS incidence - Number of vaccinated individuals per CRS case averted - Economic evaluation measure
Vynnycky_2016 [44]	- scenario 1: routine vaccination at 9 months (M & F); - scenario 2: catch-up campaign for children 9 months–14years, followed by routine vaccination at 9 months (M & F); - scenario 3: catch-up campaign for women aged 15–35 years, followed by routine vaccination at 9 months (M & F); - scenario 4: catch-up campaign at 9 months–14 years (M & F) + 15–35 years (F), followed by routine vaccine at 9 months (M & F).	Vietnam	Western Pacific	Lower-middle Income	No	37 years	MSEIRV	- Trends in CRS incidence - Number of CRS cases averted
Wesolowski_2016 [18]	- scenario 1: no vaccination; - scenario 2: routine vaccination at 9 months only (M & F); - scenario 3: routine vaccination and a start-up campaign for 9 months–10 years, followed by campaigns at 4 year intervals targeting 1–5 year-olds (M & F); - scenario 4: routine vaccination and a start-up campaign for 9 months–15 years, followed by campaigns at 4 year intervals targeting aged 1–5 year-olds (M & F); - scenario 5: routine vaccination and a start-up campaign for 9 months–20 years, followed by campaigns at 4 year intervals targeting 1–5 year-olds (M & F); - scenario 6: routine vaccination and a start-up campaign for 9 months–25 years, followed by campaigns at 4 year intervals targeting 1–5 year-olds (M & F);	Madagascar	Africa	Lowe-middle income	No	30 years	MSIRV	- Trends in rubella incidence - Trends in CRS incidence
Winter_2017 [19]	- scenario 1: no vaccine; - scenario 2: private-sector vaccine at 9–15 months and 4–6 years (M & F); - scenario 3: private sector + catch-up for children aged 9 months to 14 years + routine vaccination at 9–12 m and 16–24 m (M & F).	India	South-East Asia	Lower-middle income	Yes	30 years	MSIRV	- Trends in rubella incidence - Trends in CRS incidence
Wu_2016 [45]	- Scenario 1: routine vaccination at 9 months (M & F); - Scenario 2: routine vaccination at 6 years (M & F); - Scenario 3: routine vaccination at 9 months and 6 years (M & F); - Scenarios 4: routine vaccination at 9 months and 6 years (M & F) + catch-up for 9 months–5 years; - Scenarios 5: routine vaccination at 9 months and 6 years (M & F) + catch-up for 9 months–14 years; - Scenarios 6: routine vaccination at 9 months and 6 years (M & F) + catch-up for 9 months–39 years; - Scenario 7: routine vaccination of adolescent girls aged 12 years	Indonesia, East Java province	South-East Asia	Upper-middle income	No	50 years	SEIRV	- Trends in rubella incidence - Trends in CRS incidence - Economic evaluation measure

F: females, M: males, SEIRV: susceptible-exposed-infected-recovered-vaccinated, MSIRV: maternal immunity-susceptible-infected-recovered-vaccinated, MSEIRV: maternal immunity-susceptible-exposed-infected-recovered-vaccinated.

## Data Availability

No new data were created or analyzed in this study. Data sharing is not applicable to this article.

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
