# Peer review of "The Impact of Rubella Vaccine Introduction on Rubella Infection and Congenital Rubella Syndrome: A Systematic Review of Mathematical Modelling Studies"

_vaccines, 2021, doi:10.3390/vaccines9020084_

Round 1

Reviewer 1 Report

Rubella is a contagious viral infection. Although the infection may cause mild symptoms or even no symptoms in most people, it can cause serious problems for unborn babies whose mothers become infected during pregnancy. Rubella vaccines is the most effective method to prevent rubella and congenital rubella syndrome.  The authors of this manuscript attempt to synthesize the evidence from mathematical models regarding the impact of the introducing of rubella-containing vaccines. However as the authors mentioned that they could not synthesize the data due to the differences among the data from 7 selected studies. There is no new information was provided in this manuscript, except the evaluation of the models and summary of results from seven selected articles.

Author Response

Reviewer 1

  1. Rubella is a contagious viral infection. Although the infection may cause mild symptoms or even no symptoms in most people, it can cause serious problems for unborn babies whose mothers become infected during pregnancy. Rubella vaccines is the most effective method to prevent rubella and congenital rubella syndrome. The authors of this manuscript attempt to synthesize the evidence from mathematical models regarding the impact of the introducing of rubella-containing vaccines. However as the authors mentioned that they could not synthesize the data due to the differences among the data from 7 selected studies. There is no new information was provided in this manuscript, except the evaluation of the models and summary of results from seven selected articles.

Response: Thank you very much for your feedback. You are correct in mentioning that we were unable to synthesize the data. We think that although meta-analysis was not possible, our study contributes valuable knowledge to the field of rubella vaccination. The discrepancies in reported outcomes between the included studies can inform the methodology of future mathematical modelling studies such that outcomes are reported in a consistent manner, as far as possible, to allow for data synthesis. The purpose of a systematic review is not to generate new information, but to synthesis existing information and to increase precision of estimates when pooling is possible.

Reviewer 2 Report

Thank you for your manuscript. I have a few comments.

Abstract:

  1. line 30, do you mean rubella vaccines (which are now only included in the MMR vaccine)?
  2. you should state the checklist you used (PRISM) and the databases searched in your methods
  3. you mention that the studies were of high quality but much data seems to be missing which suggests that the checklists for reporting of economic analysis, cost-effectiveness or epidemiological studies were not followed so I wonder if this was included in your quality assessment?
  4. Your conclusions are not borne out by your results as currently presented, worth rephrasing them so that they align with what you found.

Introduction

I am not convinced of the need for the first paragraph as your SR does not describe how to undertake a mathematical systematic modelling review, for which you could use PRISMA guidelines, but seeks to answer a specific question about rubella vaccines.

The introduction is somewhere between a commentary and an introduction to your specific systematic review. It might be better to consider two papers, rather than one. It also appears from the introduction that reviews by region have already occured. Given the huge differences in regions, vaccination rates and campaigns you havent made the case for why you would want to undertake an SR?

methods

Search strategy - you do not decribe your search terms in the supplementary data, you could link to that here.

You also do not describe how you abstracted information on cost-effectiveness. There are different checklists for CEAs that you might find useful.

I could not see dates of your search or language barriers.

Did you consider looking at the WHOLIS database as there may be additional reports there?

Results

I would suggest combining reports where the age range and CRS incidence are comparable, for instance where age around 1yr vs adolescent vaccination are reported.

If you include the economic synthesis this would be better put into the context of what the actual price per DALY is as per WHO.

Discussion

I think a more thorough explanation of the limitations of the review, including the fact that it may not be possible to synthesise such data needs to be included.

Figures

Figure 1 - your figure doesnt contain reasons for exclusion (as per PRISMA guidelines), these shuold be added then you wont need the exclusion paragraph in the text.

Figure 2 - requires a legend to explain the bars

Results

I dont think you need the paragraph on excluded studies as you describe this in your consort figure 1.

Author Response

Reviewer 2

Abstract:

  1. In line 30, do you mean rubella vaccines (which are now only included in the MMR vaccine)?

Response: Thank you for the comment. Yes, we are referring to vaccines that contain rubella. There are several existing combinations that are used in different countries including measles+rubella (MR), measles, mumps +rubella (MMR), and measles+mumps+rubella+varicella (MMRV).

  1. You should state the checklist you used (PRISM) and the databases searched in your methods

Response: Thank you for the comment. In the methods section (lines 128-130) we mention that systematic review methods for classical epidemiological studies and reporting (PRISMA) guidelines were followed as far as possible.

In lines 153-158, we describe our search strategy. A comprehensive search strategy was used to identify published and unpublished studies with no language restrictions in Scopus and Pubmed databases.

  1. You mention that the studies were of high quality but much data seems to be missing which suggests that the checklists for reporting of economic analysis, cost-effectiveness or epidemiological studies were not followed so I wonder if this was included in your quality assessment?

Response: Thank you for bringing up this important point. Mathematical modelling studies do not have checklist for reporting (i.e. economical or epidemiological studies), hence we used the risk of bias assessment tool used by Harris et al. (supplementary 2). There is likely room for improvement regarding the risk of bias tools used for mathematical modelling studies, especially when economic analyses are done.

  1. Your conclusions are not borne out by your results as currently presented, worth rephrasing them so that they align with what you found.

Response: Thank you for pointing this out. We have amended the conclusion to highlight the fact that data synthesis was not possible and we provided a qualitative synthesis of study results.

Introduction

  1. I am not convinced of the need for the first paragraph as your SR does not describe how to undertake a mathematical systematic modelling review, for which you could use PRISMA guidelines, but seeks to answer a specific question about rubella vaccines.

Response: Thank you for your feedback. The approach to carrying out a systematic review of mathematical modelling studies is similar to reviews of classical epidemiology studies. We therefore carried out this systematic review of mathematical modelling studies, building on the same principles of systematic review methodology, to answer the question on rubella vaccines. We think that the sequence of ideas could be improved and we have amended the order in which the information is presented. The first paragraph has been moved to later in the introduction.

  1. The introduction is somewhere between a commentary and an introduction to your specific systematic review. It might be better to consider two papers, rather than one. It also appears from the introduction that reviews by region have already occured. Given the huge differences in regions, vaccination rates and campaigns you havent made the case for why you would want to undertake an SR?

Response: Thank you for your comment. To the best of our knowledge, there are no systematic reviews of mathematical modelling studies addressing rubella vaccine introduction. The results of individual studies were used to inform policy in several countries that were planning vaccine introduction. We mentioned in page 3, lines 117-120, that  “Given the variety of settings and modelling approaches used to evaluate the impact of RCVs, it is important to comprehensively summarize the evidence to inform policy-makers in countries that have not yet introduced RCVs, or guide adjustment of vaccination strategies where RCV are already being used.“

We think that our study question has been addressed in a single study and that we have provided a comprehensive synthesis of currently available data.

Methods

  1. Search strategy - you do not decribe your search terms in the supplementary data, you could link to that here.

Response: Thank you for your comment. We provided a description of our search strategy on page 4, lines 153-159 and the detailed search strategy is presented in supplement 1.

  1. You also do not describe how you abstracted information on cost-effectiveness. There are different checklists for CEAs that you might find useful.

Response: Thank you for your feedback. We think that checklists for specific study designs are used by authors of the individual studies. In this review, we extracted information on study characteristics, methodological quality (which included the risk of bias assessment presented in supplement 2), and outcomes as reported by the studies.

  1. I could not see dates of your search or language barriers.

Response: Thank you for your comment. We mentioned that we did not apply any language barrier (line 158, page 4).

  1. Did you consider looking at the WHOLIS database as there may be additional reports there?

Response: Thank you for the suggestion. We did not search the WHOLIS database when conducting the review. We searched this database on the 11th of January 2021 and did not find any eligible study. However, we believe our search strategy was comprehensive.

Results

  1. I would suggest combining reports where the age range and CRS incidence are comparable, for instance where age around 1yr vs adolescent vaccination are reported.

Response: thank you for the suggestion. We thought it would be easier for the reader to follow if we report the results per outcome and the included studies that reported on that outcome.

  1. If you include the economic synthesis this would be better put into the context of what the actual price per DALY is as per WHO.

Response: Thank you for the feedback. We reported economic outcomes as they were provided in the included studies. We did not perform data synthesis.

Discussion

  1. I think a more thorough explanation of the limitations of the review, including the fact that it may not be possible to synthesise such data needs to be included.

Response: Thank you for the comment. We have mentioned the reasons why we could not perform data synthesis in the results (page 5, lines 205-206), the discussion (page 15, line 46 & page 16, lines 102-104) and the conclusion (page 16, line 114).

Figures

  1. Figure 1 - your figure doesnt contain reasons for exclusion (as per PRISMA guidelines), these should be added then you wont need the exclusion paragraph in the text.

Response: Thank you for the comment. We have added the reasons for exclusion to figure 1. We also provided details on reasons for exclusion supplement 4.

  1. Figure 2 - requires a legend to explain the bars

Response: Thank you for the comment. We have added footnotes explaining the legends in the figures.

Results

  1. I dont think you need the paragraph on excluded studies as you describe this in your consort figure 1.

Response: Thank you for your feedback. We have removed this paragraph.

Reviewer 3 Report

This is a systematic review of evidence from seven mathematical models regarding the impact of introducing rubella/containing vaccines. This is a timely and relevant example of an SR of mathematical models that highlights the need to report and document modelling to become useful as evidence.

The paper is well written and easy to follow.

Major comments

The authors planned to consider several model outputs: time to rubella and congenital rubella syndrome elimination, trends in incidence of rubella and CRS, number of vaccinated individuals per CRS case averted and cost-effectiveness of vaccine introduction strategies. The planned to do a meta-analysis using statistical modelling of modelling outputs. My main concern is the authors’ decision to instead to do a qualitative narrative “given that data synthesis was not feasible”. I have two issues with this.

  1. Quality of data and exploration of uncertainty were in the RoB-analysis judged to be fully addressed or partially addressed (according to definitions in tables in supplementary 2). At the same time, the authors conclude on page 5 row 203 that several of the included studies lacked an uncertainty assessment. How can some studies both lack an uncertainty assessment and partially address uncertainty? This points to a possible limitation in the RoB analysis that considers data and uncertainty assessment by one single criterion (see comment below 4).
  2. The advantage of modelling over experimental studies is that one can rerun the modelling at a low cost. Instead of abandoning the plan, one option would have been to use the models for the purpose of this SR. Why was it not possible to extract the outputs of interest from the published models? I recommend the authors to at least reflect on the possibility of using the published models for quantitative synthesis.

My second concern is the risk of bias analysis. Risk of bias was evaluated with a scheme consisting of 14 criteria developed and used by several previous studies. RoB-analysis for models is a relatively new phenomenon, and I am not 100% confident in the scheme that is used. For experimental evidence, bias is about systematic or random errors associated with the estimated effect or the evidence for or against a hypothesis or causal relationship.

  1. What is the meaning of bias in the context of modelling?
  2. The 14 criteria constitute a mixture of different things such as reporting, model definition, methods for model calibration and validation, quality of input data and uncertainty assessment. I know this paper is not suggesting an RoB analysis, but I welcome a critical discussion of the RoB analysis. For example, data quality and uncertainty are evaluated together on a three-point scale, while this is a major point of the model summarizing several of the other points. How can to what extent a model has a clear aim be given the same weight if it considers uncertainties? How is bias related to the reporting of the model? Some criteria justify that the model is modelling what we want to model or justify the reliability of the model. Which criteria are about reproducibility? In general, for models to count as evidence, they should be available for use to be able to produce the outputs of interest (see my comment 1).

Minor comments

  1. The introduction begins with two paragraphs about systematic reviews and the use of SRs of evidence from mathematical modelling. I find this odd because the main aim of the study is to summarise (modelling) evidence about vaccine introduction strategies, where SR is the method to achieve the aim. The paper would be easier to read if the introduction started with paragraph 3 and 4, followed by what is now 1 and 2.
  2. Why was IBMs excluded?
  3. Page 14 Line 45 All models were deterministic and age-structured are mentioned in the same sentence that the results did not allow for data synthesis. Is there a link between these two things? If not, I recommend placing them in two separate sentences. What are the possible implications of all models being deterministic age-structured? Is it a limitation of the findings that there were only deterministic models?
  4. Page 14 row 47. The finding that “low coverage with rubella vaccines led to higher CRS incidence compared to no vaccination” is counter-intuitive. Is there an explanation for this finding?
  5. Page 14 row 48. Why is the finding that “strategies involving routine vaccination of children at specific ages outside the routine infant dose were more cost-effective than strategies involving mass campaigns” seen as interesting? Was this against, what is expected? Do the models say anything about how much better one strategy is compared to another?

Author Response

Reviewer 3

This is a systematic review of evidence from seven mathematical models regarding the impact of introducing rubella/containing vaccines. This is a timely and relevant example of an SR of mathematical models that highlights the need to report and document modelling to become useful as evidence.

The paper is well written and easy to follow.

Major comments

  1. The authors planned to consider several model outputs: time to rubella and congenital rubella syndrome elimination, trends in incidence of rubella and CRS, number of vaccinated individuals per CRS case averted and cost-effectiveness of vaccine introduction strategies. The planned to do a meta-analysis using statistical modelling of modelling outputs. My main concern is the authors’ decision to instead to do a qualitative narrative “given that data synthesis was not feasible”. I have two issues with this.

Quality of data and exploration of uncertainty were in the RoB-analysis judged to be fully addressed or partially addressed (according to definitions in tables in supplementary 2). At the same time, the authors conclude on page 5 row 203 that several of the included studies lacked an uncertainty assessment. How can some studies both lack an uncertainty assessment and partially address uncertainty? This points to a possible limitation in the RoB analysis that considers data and uncertainty assessment by one single criterion (see comment below 4).

Response: Thank you for your feedback. The risk of bias item pertaining to exploration of uncertainty is referring to uncertainty in the model structure, parameters and/or assumptions. A meta-analysis was not possible because of the lack of uncertainty intervals in the reported outcomes of several included studies. We have amended the sentence in line 203 of page five to: “Only a narrative synthesis was done because several included studies did not estimate uncertainty intervals when reporting outcomes”. The estimate and its level of precision are required for meta-analyses in order to allocate the appropriate weights.

  1. The advantage of modelling over experimental studies is that one can rerun the modelling at a low cost. Instead of abandoning the plan, one option would have been to use the models for the purpose of this SR. Why was it not possible to extract the outputs of interest from the published models? I recommend the authors to at least reflect on the possibility of using the published models for quantitative synthesis.

Response: Thank you for your comment. We agree that unlike experimental studies, simulations in mathematical models can be rerun. While it is possible to obtain the code of the published models, these models were built by different researchers with distinct approaches using diverse programming languages. The requirements in terms of time and capacity to rerun the models in the included studies, while adapting them to obtain outputs that are similar enough for data synthesis, are enormous. However, our findings can guide current and future modelling studies to avoid this limitation in future systematic reviews of mathematical modelling studies.

3.My second concern is the risk of bias analysis. Risk of bias was evaluated with a scheme consisting of 14 criteria developed and used by several previous studies. RoB-analysis for models is a relatively new phenomenon, and I am not 100% confident in the scheme that is used. For experimental evidence, bias is about systematic or random errors associated with the estimated effect or the evidence for or against a hypothesis or causal relationship.

What is the meaning of bias in the context of modelling?

The 14 criteria constitute a mixture of different things such as reporting, model definition, methods for model calibration and validation, quality of input data and uncertainty assessment. I know this paper is not suggesting an RoB analysis, but I welcome a critical discussion of the RoB analysis. For example, data quality and uncertainty are evaluated together on a three-point scale, while this is a major point of the model summarizing several of the other points. How can to what extent a model has a clear aim be given the same weight if it considers uncertainties? How is bias related to the reporting of the model? Some criteria justify that the model is modelling what we want to model or justify the reliability of the model. Which criteria are about reproducibility? In general, for models to count as evidence, they should be available for use to be able to produce the outputs of interest (see my comment 1).

Response: Thank you for pointing out this important issue. We agree that risk of bias assessment is challenging and that the risk of bias tool used in our paper could be discussed further. In the sixth paragraph of the discussion, we have further developed possible limitations of the risk of bias tool; lines 89-92, page 15.

Minor comments

4.The introduction begins with two paragraphs about systematic reviews and the use of SRs of evidence from mathematical modelling. I find this odd because the main aim of the study is to summarise (modelling) evidence about vaccine introduction strategies, where SR is the method to achieve the aim. The paper would be easier to read if the introduction started with paragraph 3 and 4, followed by what is now 1 and 2.

Response: Thank you for pointing this out. We have rearrange the introduction by swapping paragraphs 3 and 4 with paragraphs 1 and 2.

5.Why was IBMs excluded?

Response: Thank you for the question. We did not deliberately exclude individual-based models from this review. Our search did not retrieve any study that used an IBM approach so no study was excluded for that reason. We have deleted the phrase in the abstract that indicated that we were excluding IBMs.

6.Page 14 Line 45 All models were deterministic and age-structured are mentioned in the same sentence that the results did not allow for data synthesis. Is there a link between these two things? If not, I recommend placing them in two separate sentences. What are the possible implications of all models being deterministic age-structured? Is it a limitation of the findings that there were only deterministic models?

Response: Thank you for your comment. We have split the sentence into two different sentences.

There are no specific implications for all the models being deterministic age structured and this does not represent a limitation. Several modelling groups have explored different ways to model rubella transmission and the deterministic age-structured approach is considered by many authors to be robust.

7.Page 14 row 47. The finding that “low coverage with rubella vaccines led to higher CRS incidence compared to no vaccination” is counter-intuitive. Is there an explanation for this finding?

Response: Thank you for your feedback. Introduction of rubella vaccine at low coverage has been documented (Costa Rica and Greece) to cause transient increases in incidence of congenital rubella syndrome and we included this explanation in the second paragraph of the discussion on page 14, lines 52-60.

8.Page 14 row 48. Why is the finding that “strategies involving routine vaccination of children at specific ages outside the routine infant dose were more cost-effective than strategies involving mass campaigns” seen as interesting? Was this against, what is expected? Do the models say anything about how much better one strategy is compared to another?

Response: Thank you for your comment. This finding is interesting because rubella vaccines are usually given as routine vaccination in infants and mass campaigns targeting wide age ranges.

The only included studies that reported vaccine costs per CRS case averted (Wu et al. and Motaze et al.) found lower costs in scenarios that entailed vaccination of specific ages outside the routine infant dose compared to scenarios that included mass campaigns. These results are presented in lines 36-41, pages 13 and 14.

Round 2

Reviewer 1 Report

The authors addressed comments from reviewers; the revised manuscript is more logically clear.

Although they failed to draw broad conclusions on the impact of the rubella vaccine on rubella virus infection or congenital rubella syndrome, the authors evaluated the quality of mathematical modeling used in the selected studies. The results provide a guideline for future mathematical modeling studies.

However, the title would be more accurate, if the authors replace ”rubella infection” with “rubella virus infection”.

Reviewer 2 Report

Thank you for your revised manuscript. I have no further comments

Reviewer 3 Report

I find that the authors have adequately adressed my comments.